Resource

# Development of K-CORE: a web-based platform for integrated clinico-genomic analysis

Juyeon Hwang[1,*] , Jae Woo Ahn[1,*], Jae Wook Lee[2], So-Youn Jung[4,7], Eun-Gyeong Lee[4,7], Heejung Chae[11], Harim Koo[9], Hyosoung Cha[1,6,8], Junetae Kim[6,8], Kwangmin Kim[12] , Dongwoo Lee[10], Junhyung Park[12], Sun-Young Kong[3,5,7] , Kui Son Choi[1,6] , Hyun-Jin Kim[1,6] 

One of the key challenges in cancer treatment and precision oncology is the use of multi-omics data and their integration into matched clinical information. Although several analytical portals have been developed, most platforms do not support user-uploaded data or the integrated analysis of clinical and multi-omics datasets. To address these limitations, we developed the Korea Cancer Omics Research (K-CORE) portal, a user-friendly analytical platform designed to integrate and analyze multi-omics and clinical data. K-CORE supports various omics levels and a wide range of analytical tools. To validate the utility and reproducibility of the K-CORE, we designed synthetic datasets that reflected real-world omics data distributions. The analytical results from K-CORE were compared side by side with those from widely used R packages such as maftools and edgeR. In conclusion, K-CORE offers a practical and intuitive platform for the multidomain integration of clinical and omics data, supporting the advancement of precision oncology. Nevertheless, as analytical technologies and precision oncology continue to evolve, continuous maintenance and user feedback will become essential for future platform improvements.

## Introduction

The importance of precision medicine for cancer treatment has increased (Collins & Varmus, 2015). The most important aspect of current precision medicine is the comprehensive use of omics data and clinical information to facilitate precise patient stratification, diagnosis, and identification of candidate therapeutic genes

(Razzaghdoust et al, 2021; Kim et al, 2024). In parallel with these research trends, precision oncology studies have shifted from single-omics (e.g., genomics or transcriptomics) to multi-omics approaches. This paradigm suggests novel perspectives for precision oncology by enabling a more holistic understanding of disease mechanisms and treatment response (Kim et al, 2024). Furthermore, as the importance of omics data has been emphasized, clinical information has become a crucial component of precision oncology research. The clinical information of patients with cancer (e.g., survival ratio and prognosis variations by anticancer drugs) is recognized as important as omics data for advancing medical research (Choi et al, 2023).

With the advancement of precision oncology, the global accumulation of large-scale omics datasets has accelerated (e.g., The Cancer Genome Atlas, TCGA) (Cancer Genome Atlas Research Network et al, 2013). Comprehensive data portals such as TCGA and the Clinical Proteomic Tumor Analysis Consortium provide multidimensional datasets encompassing DNA, RNA, and protein levels. These publicly available databases hold a significant promise for driving progress in cancer research. However, the effective use of omics data requires substantial computational and bioinformatics expertise, which remains a major bottleneck because of the limited availability of trained professionals.

In parallel with analytical limitations, several data portals, such as cBioPortal and LinkedOmics, provide integrated visualization and analysis functionalities (Cerami et al, 2012; Vasaikar et al, 2018) (Table 1). In addition, several analytical platforms and packages (e.g., Galaxy and TCGAnalyzeR) have been developed for the analysis of omics data (Galaxy Community, 2024; Zengin et al, 2024) (Table 1). Nevertheless, currently available analytical portals for precision medicine remain limited in terms of accessibility, flexibility, and integrative functionality. Most platforms do not support

[1]National Cancer Control Institute, National Cancer Center, Goyang, South Korea   [2]Nephrology Clinic, National Cancer Center, Goyang, South Korea   [3]Department of Laboratory Medicine, National Cancer Center, Goyang, South Korea   [4]Department of Surgery, Center for Breast Cancer, Hospital, National Cancer Center, Goyang, South Korea   [5]Targeted Therapy Branch, National Cancer Center, Goyang, South Korea   [6]Department of Public Health & AI, Graduate School of Cancer Science and Policy, National Cancer Center, Goyang, South Korea   [7]Department of Cancer Biomedical Science, Graduate School of Cancer Science and Policy, National Cancer Center, Goyang, South Korea   [8]Healthcare AI Team, National Cancer Center, Goyang, South Korea   [9]Department of Medical Science Convergence, Graduate School of Medical Science, University of Ulsan, Ulsan, Republic of Korea   [10]Hancom Carelink Inc., Seoul, Republic of Korea   [11]Department of Hematology and Medical Oncology, Comprehensive Cancer Center, Seoul National University Bundang Hospital, Seongnam, South Korea   [12]3BIGS Co., Ltd., Hwaseong, South Korea

Correspondence: ksy@ncc.re.kr; kschoi@ncc.re.kr; hyunjin@ncc.re.kr
*Juyeon Hwang and Jae Woo Ahn contributed equally to this work

**Table 1. Task-based comparison of K-CORE with existing omics analysis portals.**

| Platform (official website) | Developed by | Supported data | Visualization for user-uploaded data | Direct comparison with K-CORE |
|---|---|---|---|---|
| cBioPortal (https://www.cbioportal.org) | Memorial Sloan Kettering Cancer Center | Public and user-uploaded data | OncoPrint and lollipop plot | Optimized for analyzing large-scale public genomic datasets (e.g., TCGA). Supports user-uploaded data with limited visualization (OncoPrint and lollipop plot), and multi-omics integration is not available. Local hosting is required for extended features, which limits usability for nontechnical users |
| UCSC Xena (https://xena.ucsc.edu) | University of California, Santa Cruz | Public and user-uploaded data | Kaplan–Meier analysis, box plot, and scatter plot | Supports public and user-uploaded data, enabling visualization of genomic and clinical information via the Xena browser. Private data analysis requires installing a local Xena hub, which may be challenging for nontechnical users |
| Galaxy (https://usegalaxy.org) | Pennsylvania State University and Johns Hopkins University | Public and user-uploaded data | Line track, feature track, box plot, histogram, scatter plot, and pie chart | Provides customizable workflows for analyzing various types of omics data. Although basic visualization is supported through its built-in tools, the platform does not support integrated views of multiple omics types or connections to clinical information |
| TCGAnalyzeR (http://tcganalyzer.mu.edu.tr) | Mugla Sitki Kocman University | Public | Not supported | Designed for TCGA-based genomic analysis, providing subcohort comparisons and filtering based on specific genes or patient groups. However, multi-omics data integration and user data upload are not supported |
| LinkedOmics (https://www.linkedomics.org) | Baylor College of Medicine | Public | Not supported | Provides multi-omics data analysis and network-based interpretation using TCGA and CPTAC data. However, user data upload and customized analysis are not supported |
| K-CORE (https://www.cancerdata.re.kr/k-core) | National Cancer Center, South Korea | User-uploaded data (multi-omics and clinical data) | OncoPrint, Circos plot, volcano plot, heatmap, box plot, survival curve, Cox regression, PCA, fusion plot, CNV plot, correlation plot, variant summary, and lollipop chart | Supports web-based analysis of user-uploaded clinical and multi-omics datasets with guided QC, dynamic activation of compatible modules, clinically driven sample filtering, and integrated visualization without local installation |

Representative omics data analysis portals were compared according to developer, supported data types, visualization functions for user-uploaded data, and key differences from K-CORE. K-CORE supports integrated web-based analysis of user-uploaded clinical and multi-omics datasets through guided quality control, dynamic module activation, clinical filtering, and visualization without local installation.

the analysis of user-uploaded data or the comprehensive integration of multi-omics data with clinical information.

To overcome these limitations, the Korea Cancer Omics Research (K-CORE) portal was developed as a user-friendly analytical platform designed to integrate and analyze multi-omics and clinical datasets (https://www.cancerdata.re.kr/k-core). K-CORE offers accessible and efficient analytical tools for exploring not only complex biological data, such as genomic alterations, DNA methylation, and RNA and protein expression profiles, but also clinical information related to cancer. By improving accessibility to analytical tools, simplifying data processing and visualization, and enhancing interpretability, K-CORE addresses the key limitations of existing public platforms and facilitates broader applications of precision oncology research.

# Results

## Overall workflow of the platform

The K-CORE platform is structured around three main steps:

(1) Uploading clinical and multi-omics datasets.
(2) Data filtering based on clinical characteristics or gene selection.
(3) Generating interactive visualization and analysis results (Fig 1).

K-CORE supports upload of clinical data together with gene mutation, copy-number variation (CNV), DNA methylation, RNA expression, protein expression, gene fusion, and phosphorylation files. Key navigation, upload, and visualization pages are available in English via the portal language toggle (Fig S1). During upload, data type–specific help dialogs summarize required columns and provide sample TSV templates (Fig S2A and B). In addition, an English user manual is now available through the customer-support menu (Fig S3).

Upon upload, the platform performs automated QC to validate file structure, required fields, value types, and sample ID consistency across datasets (Fig S4). Erroneous rows are flagged before analysis. In the multidata workflow, a clinical information file is required and projects are stored under authenticated accounts, whereas single-data analyses remain ephemeral.

After QC, users can refine their datasets by applying filters based on clinical variables (e.g., age, sex, and smoking status), predefined cancer-associated gene sets, or user-defined gene sets. K-CORE then activates only the analysis modules compatible with the uploaded data types. Depending on dataset composition, up to 13 visualization options are available, including variant summary, OncoPrint, volcano plot, heatmap, survival curve, correlation plot, fusion plot, Sankey plot, Circos plot, lollipop chart, CNV visualization, box plot, and PCA.

## Interactive visualization features of the platform

K-CORE supports multi-omics landscape visualization, mutation and expression analysis, survival analysis, and several supporting tools within a shared filtering framework. Across modules, users can interactively subset cohorts by clinical variables or gene sets, allowing the same uploaded dataset to be interrogated from complementary analytic perspectives. Reference example analyses are available across the full set of analytical modules through the example-data interface, allowing users to inspect the breadth of supported tools before uploading their own datasets (Fig S5A–C).

## Multi-omics landscape visualization using OncoPrint and Circos plots

OncoPrint provides a graphical overview of the genetic alterations across individual samples (Fig 2A). Color and shape are used to distinguish between the mutation types and expression levels. It integrates multiple data types, including seven categories of DNA and copy-number alterations, as well as RNA and protein expression. Users can customize the visualization thresholds for gene expression and copy-number alterations. Clinical variables can be annotated in the OncoPrint database using dropdown menus.

The Circos plot was used to visualize multi-omics alterations in individual samples (Fig 2B). Concentric rings represent genomic mutations, copy-number alterations, RNA and protein expression levels, DNA methylation, and structural variations. Only uploaded and available data types are dynamically displayed.

## Visualization of mutation and phosphorylation data

The variant summary provides a comprehensive overview of genomic mutations across the dataset (Fig 3A). It visualizes the distribution of mutation classifications such as missense, nonsense, and frameshift variants. It also presents the number of variants, including SNPs, insertions, and deletions. The top 10 most frequently mutated genes are displayed and categorized by mutation type. In addition, a box plot summarizes the distribution of mutation classifications across the samples.

The lollipop plot visualizes mutation or phosphorylation sites mapped onto sequences overlaid with annotated domain structures (Fig 3B). Users can select either a mutation or phosphorylation for visualization purposes. The summary table provides sample-specific information, including sample IDs, mutation types, amino acid changes, phosphorylated sites, and corresponding genes.

## Visualization of expression data

The heatmap supports the hierarchical and K-means clustering of the multi-omics data, including RNA and protein expression, DNA methylation, and phosphorylation data (Fig 4A). Users can select raw counts or normalized data for RNA and protein datasets and apply gene selection for methylation and phosphorylation data. Clinical annotations can be added to the top of the heatmap, and spectrum-range customization is available for all the clustering methods. In K-means clustering, users can define the number of clusters and choose whether clustering should be performed at the gene or sample levels.

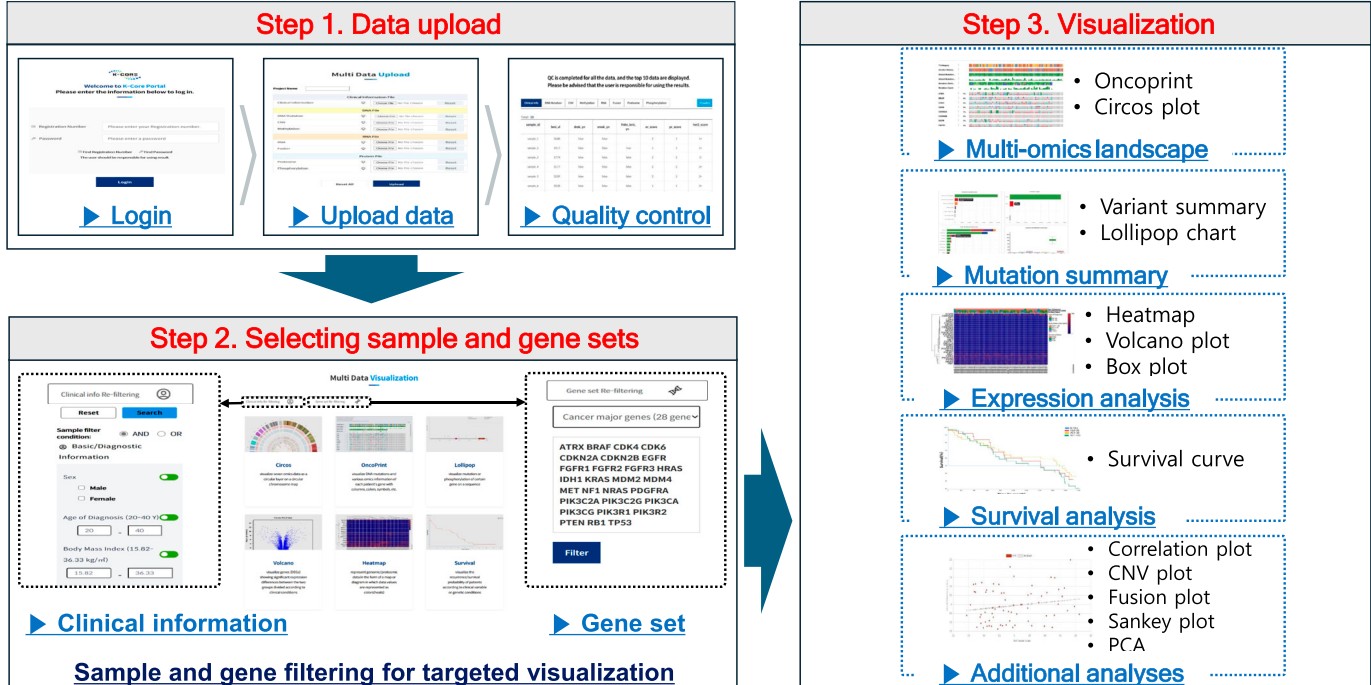

**Figure 1. Workflow of the K-CORE platform.**

The platform enables structured, stepwise analysis of clinical and multi-omics data. After uploading datasets and completing automated quality checks, users filter samples and genes using clinical attributes and predefined cancer-related gene sets. Visualization options are then automatically suggested based on data type and filtering results, covering five categories such as mutation summary, expression analysis, and additional tools.

The box plot visualizes the distribution of RNA or proteome expression values across groups, such as tumor and normal samples (Fig 4B). Users can choose between raw count and normalized data options when displaying expression values. Each box plot summarizes the range and distribution of expression values across the sample groups.

The volcano plot visualizes the differential expression based on transcriptome or proteome data (Fig 4C). Users can compare tumor and normal samples directly or flexibly define analysis groups based solely on clinical variables from the uploaded dataset. The clinical variables are automatically categorized into continuous, categorical, or Boolean types to support flexible group definitions. For continuous variables, users can define cutoff points; for categorical variables, groups are assigned by selecting categories through checkboxes; and for Boolean variables, grouping is automatically determined. Users can optionally set thresholds for the $\log_2$ fold change and $P$-values to distinguish significantly up-regulated or down-regulated expression levels. Preselected genes from the gene set panel are highlighted with black dots and labeled accordingly. A summary table listing the gene names, $\log_2$ fold changes, $P$-values, and false discovery rates corresponding to the visualized results is also provided.

## Survival analysis based on clinical and omics stratification

Survival and recurrence outcomes are visualized based on clinical and omics group classifications (Fig 5A–D). Groups can be stratified based on clinical attributes or by selecting specific genes for

omics-based classification. When using omics data, samples are grouped by DNA mutation status or RNA and protein expression levels. Clinical variables are handled according to their type; continuous variables can be stratified by setting cutoff values and categorical variables by selecting categories through checkboxes, and Boolean variables are grouped automatically. Cox regression analysis is performed separately using the clinical variables to estimate the hazards associated with the event of interest. Summary tables are provided for each analysis. Survival and recurrence outputs include time, survival probability, and sample information, whereas the Cox regression results include coefficient estimates, confidence intervals, and $P$-values.

## Other visualization features

In addition to the core features, several extended visualizations are available to support further analysis (Fig S6A–E). The correlation plot depicts the relationship between RNA and proteome expression levels for the selected genes, providing correlation coefficients and $P$-values. The CNV plots display copy-number alterations for selected genes across the genome. Fusion plot visualization allows the grouping of samples based on clinical attributes and uses a Venn diagram to show fusion genes unique to or shared between groups. Additional details, such as breakpoints and junction read counts, can be accessed through an interactive table and viewer. Furthermore, a Sankey plot was used to visualize the structured relationships of selected genes and variant types to disease and drug information. Patient summary reports provide

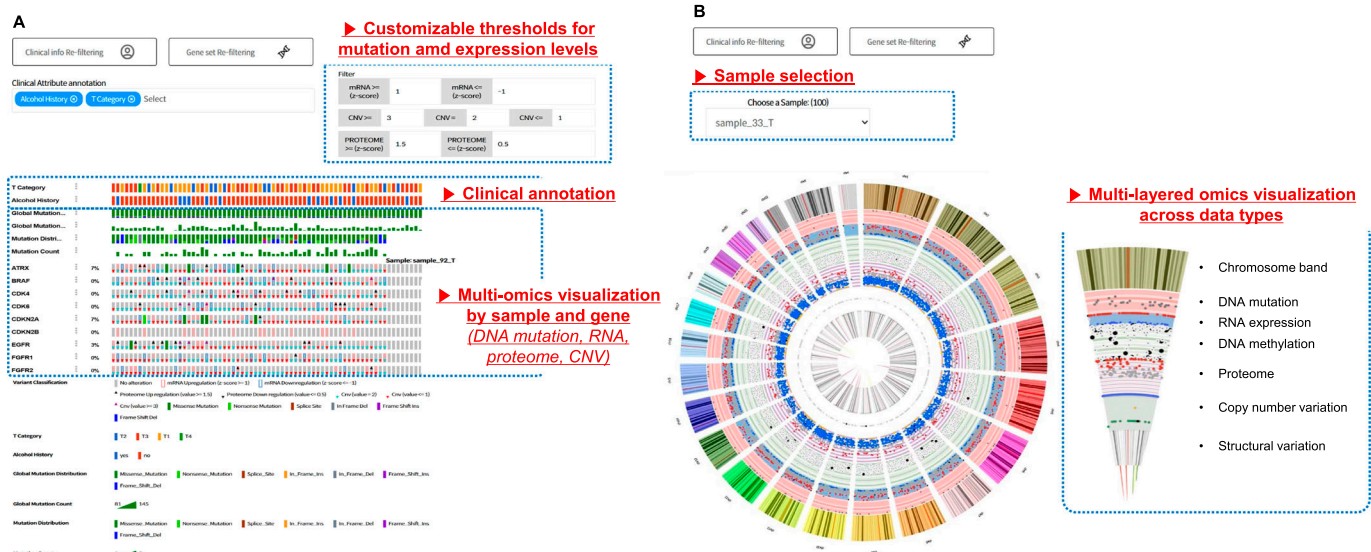

**Figure 2. Sample-level multi-omics landscape visualizations with OncoPrint and Circos plots.**
**(A, B)** OncoPrint shows gene-level multi-omics alterations across samples, combining somatic mutations, RNA and protein expression, and CNV data. Clinical annotations appear above the plot. Data types are encoded by color and shape. Users can adjust thresholds for expression and CNV to refine the view; and (B) Circos plot displays a circular view of omics alterations in a selected sample. Concentric rings represent somatic mutations, RNA expression, DNA methylation, proteomics, CNVs, and structural variants. The plot adapts to the uploaded data types available for each sample.

integrated per-sample classifications, including mutation status, expression levels, and proteome patterns. Finally, PCA visualizes the distribution of tumor and normal samples along two principal components using the selected RNA or proteome data.

### Validation of analytical results

To evaluate the accuracy and utility of the K-CORE platform, we validated its analytical functions using synthetic datasets derived from the publicly available LinkedOmics dataset (Fig S7). The validation process focused on assessing the consistency of the results generated by K-CORE compared with those obtained using widely used R packages. First, genomic mutation analysis was validated using OncoPrint, variant summary, and lollipop plots (Figs 6A and B and S8A). OncoPrint, which visualizes the mutation landscape of selected genes, showed consistent mutation frequencies for major genes such as *TP53* (57%) and *PIK3CA* (53%) across all samples. These results were consistent between the K-CORE and maftools R packages. Similarly, the variant summary and lollipop plots showed comparable distributions of mutation types and genomic positions between K-CORE and maftools. Differential expression analysis was validated using synthetic RNA expression data generated using K-CORE. Volcano plots created using K-CORE and the R package (edgeR) produced consistent results in terms of $\log_2$ fold change and statistical significance (Fig 6C). Box plots were also used to assess the distribution of gene expression levels and the consistency of statistical outcomes for individual genes (Fig S8B). Finally, the clinical data were validated through survival analysis. Survival curves were compared between smokers and nonsmokers using synthetic clinical data provided by the K-CORE. The KM survival curves generated by K-CORE and those

generated using R showed identical *P*-values (*P* = 0.0694), confirming the reproducibility and accuracy of the survival analysis module (Fig 6D).

## Discussion

The increasing availability of clinical and multi-omics data has created a growing demand for integrative and accessible tools that enable exploratory analysis without specialized programming skills. To address this challenge, K-CORE was developed as a web-based platform that allows users to upload, filter, and visualize diverse omics and clinical datasets using interactive analysis functions. K-CORE supports various data types, including clinical variables, mutations, RNA expression data, methylation, and proteomics, and provides visualization outputs such as OncoPrint, volcano plot, heatmap, and survival curve to facilitate the exploration of integrated patterns in a user-friendly environment.

Several web-based platforms, including cBioPortal, UCSC Xena, Galaxy, TCGAnalyzeR, and LinkedOmics, are widely used for omics data analysis (Table 1). However, each platform has limitations, particularly regarding integrated multi-omics analysis and support for user-uploaded datasets. For example, cBioPortal primarily focuses on public cancer genomics datasets, such as TCGA, MSKCC, and AACR GENIE (Cerami et al, 2012; Gao et al, 2013). Although it offers multidomain visualizations such as survival curves and mutation frequency maps, comprehensive multi-omics integration remains limited. User-uploaded data can only be visualized through OncoPrints and lollipop plots, and full analysis requires installing a local server. UCSC Xena enables the visualization of gene expression, CNV, DNA methylation, and clinical data

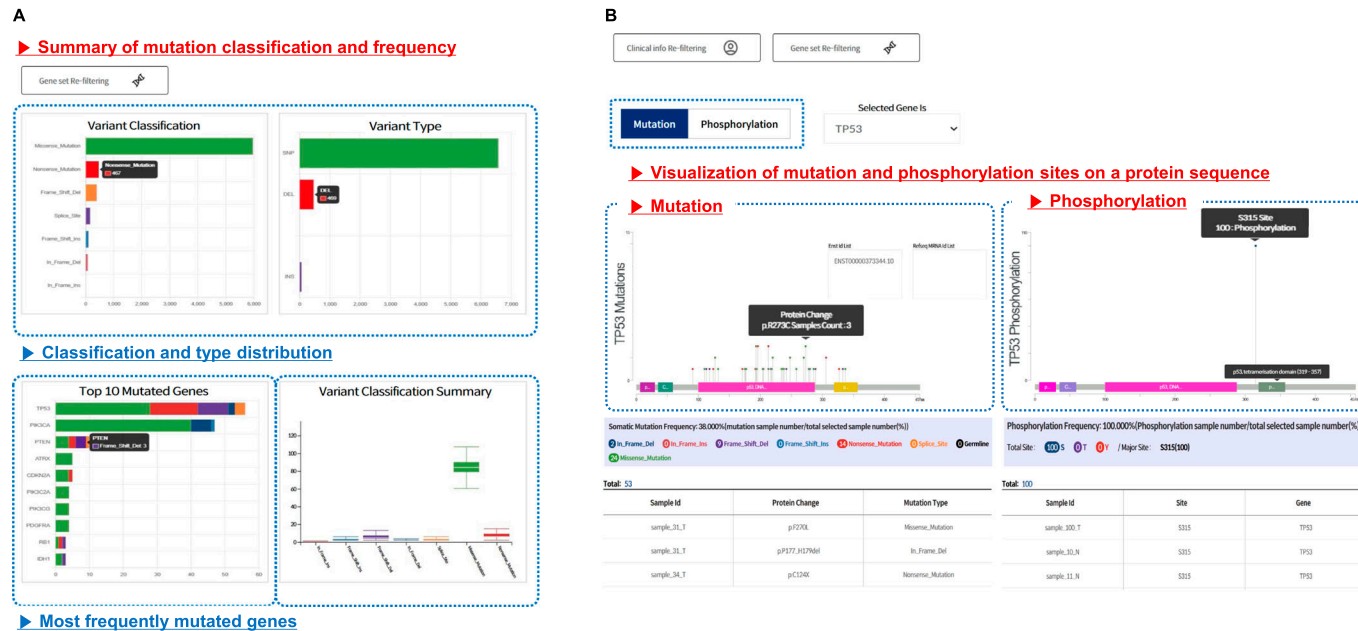

**Figure 3. Visualization of mutation patterns and phosphorylation sites using variant summary and lollipop plots.**
**(A, B)** Variant summary plots show the distribution of mutation types, the most frequently mutated genes, and per-gene classification patterns across the dataset; and (B) lollipop plots display mutation or phosphorylation sites along protein sequences, overlaid with domain structures. Sample-level annotations are shown in the accompanying table.

(Goldman et al, 2020; Perez et al, 2025) but lacks native multi-omics integration and requires manual synthesis across data types. Galaxy provides a flexible environment for building computational workflows across various omics datasets (Giardine et al, 2005; Galaxy Community, 2022, 2024) but provides limited support for integrated omics visualization and interactive exploration. TCGA-nalyzeR allows the interactive exploration of TCGA datasets through predefined analyses such as survival, mutation, and co-expression (Zengin et al, 2024) but offers little flexibility beyond fixed pipelines, and multi-omics exploration is not its primary focus. LinkedOmics supports correlation and enrichment analyses across transcriptomic, proteomic, and clinical data in 32 TCGA cancer types (Vasaikar et al, 2018); however, analyses are restricted to predefined models, limiting exploratory flexibility.

Recognizing these limitations, we designed the K-CORE to help overcome the key technical barriers present on existing platforms. It enables the seamless exploration of user-uploaded clinical and multi-omics datasets through automatically activated analysis modules, flexible filtering, and sample grouping options based on uploaded data types. The K-CORE supports exploratory analysis and assists in the development of research hypotheses.

However, this study has several limitations. The flexibility of user-defined analyses increases system complexity, necessitating ongoing backend optimization. Scalability challenges may arise with heavy concurrent usage, particularly with large-scale omics datasets and computation-intensive visualization. Future developments will address these challenges through the enhancement of server performance and adaptive workload management. In addition, we plan to integrate artificial intelligence and machine learning methods to support

outcome prediction, biomarker discovery, and subgroup identification, further extending the platform's analytical capabilities in precision oncology research (Nam et al, 2024; Nguyen & Vafaee, 2025 Preprint).

# Materials and Methods

### System configuration and operation policy

The K-CORE portal is a web-based application developed using Python v3.8 (RRID:SCR_008394) and Node.js v16.0.0. The backend was built using Python with the Django framework (RRID:SCR_012855) and Django REST API, and the front end was implemented in React.js with visualization libraries including D3.js, CanvasXpress.js, Circos.js, and Oncoprint.js. K-CORE uses PostgreSQL v13.3 (RRID:SCR_021067) as the primary database for user and application data and SQLite3 (RRID: SCR_017672) for temporary project storage during custom multidata visualization. To balance usability, storage efficiency, and protection of uploaded data, K-CORE supports both ephemeral single-data analysis and project-based multidata analysis. In the current deployment, each account can maintain up to five multidata projects simultaneously; project data are retained for 2 wk after creation and can be extended once by an additional 2 wk.

### User access, interface language, and data management

Key user-facing portal pages are available in English through a language toggle on the main site (Fig S1), including the landing

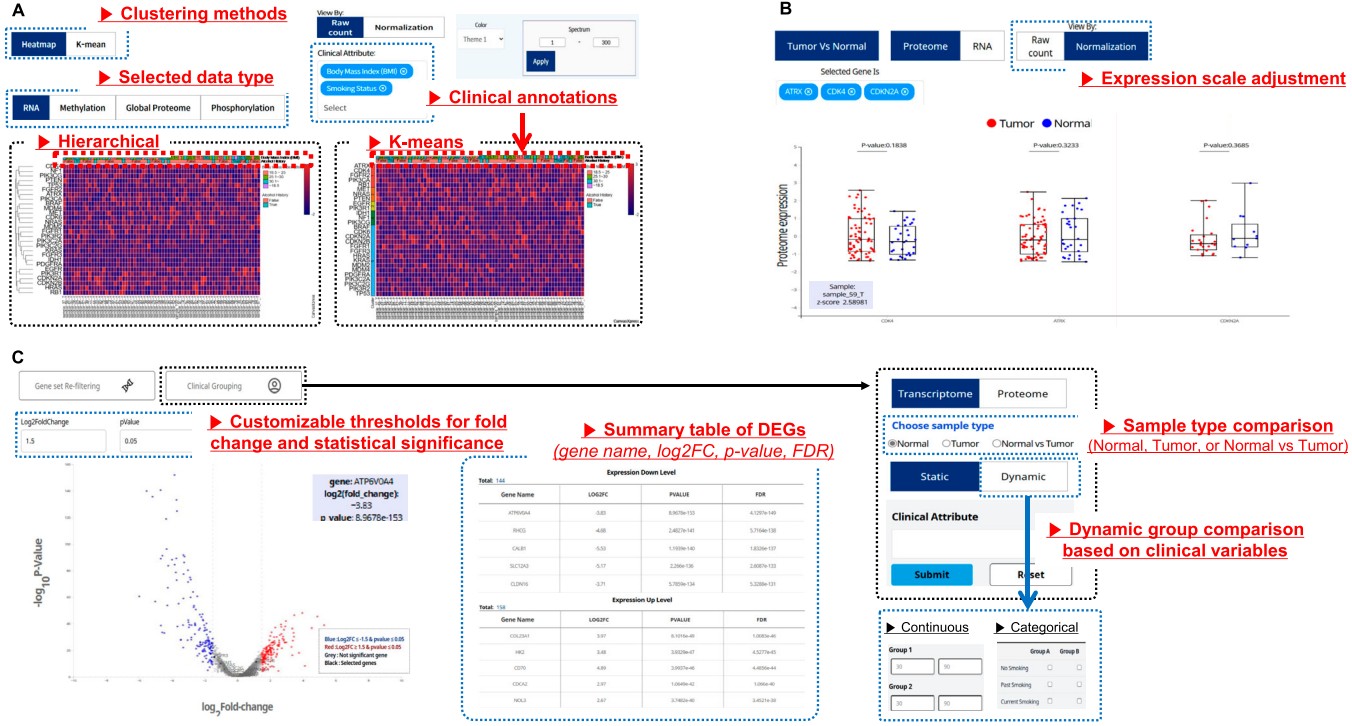

**Figure 4. Gene expression analysis using heatmap, box plot, and volcano plot.**
**(A, B, C)** Heatmaps show sample-level expression patterns using hierarchical or K-means clustering. Users can choose among four omics types (RNA, proteome, methylation, and phosphorylation), adjust expression scales, and apply clinical annotations to columns; (B) box plots display gene-level expression distributions across tumor and normal samples. Users can toggle between raw and normalized values, and compare expression levels across selected genes; and (C) volcano plots visualize differentially expressed genes based on fold-change and *P*-value thresholds. Clinical groups are defined using continuous or categorical variables, and dynamic comparisons are supported. Selected genes and summary statistics (log₂FC, *P*-value, FDR) are listed in the result table.

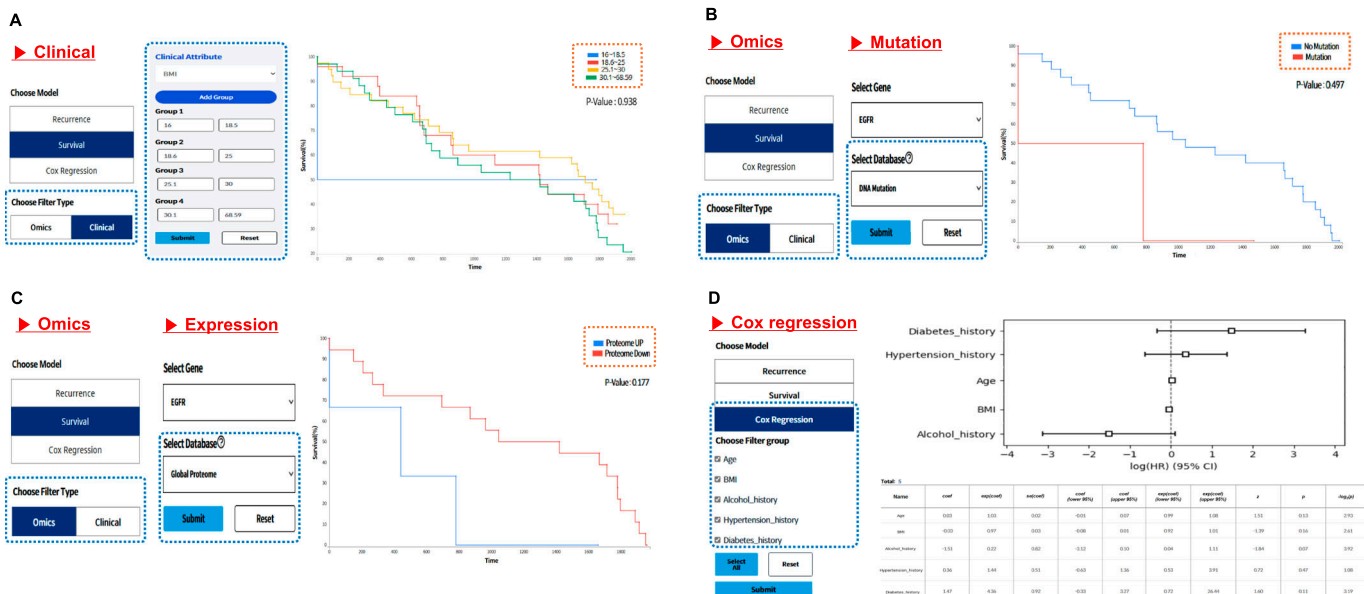

**Figure 5. Survival and Cox regression analyses based on clinical and omics-based groupings.**
**(A, B, C, D)** Survival curves based on clinical variables are generated by grouping samples using custom cutoff values. Continuous, categorical, and Boolean variables are automatically handled according to their type; (B, C) omics-based survival curves compare groups based on mutation status (e.g., *EGFR* mutated versus WT) or expression levels (e.g., high versus low RNA or proteome expression). Users can select genes and omics types to define comparison groups; and (D) Cox regression quantifies the influence of clinical variables on survival. Results are presented as a forest plot with hazard ratios, confidence intervals, and *P*-values. A summary table lists the corresponding regression coefficients.

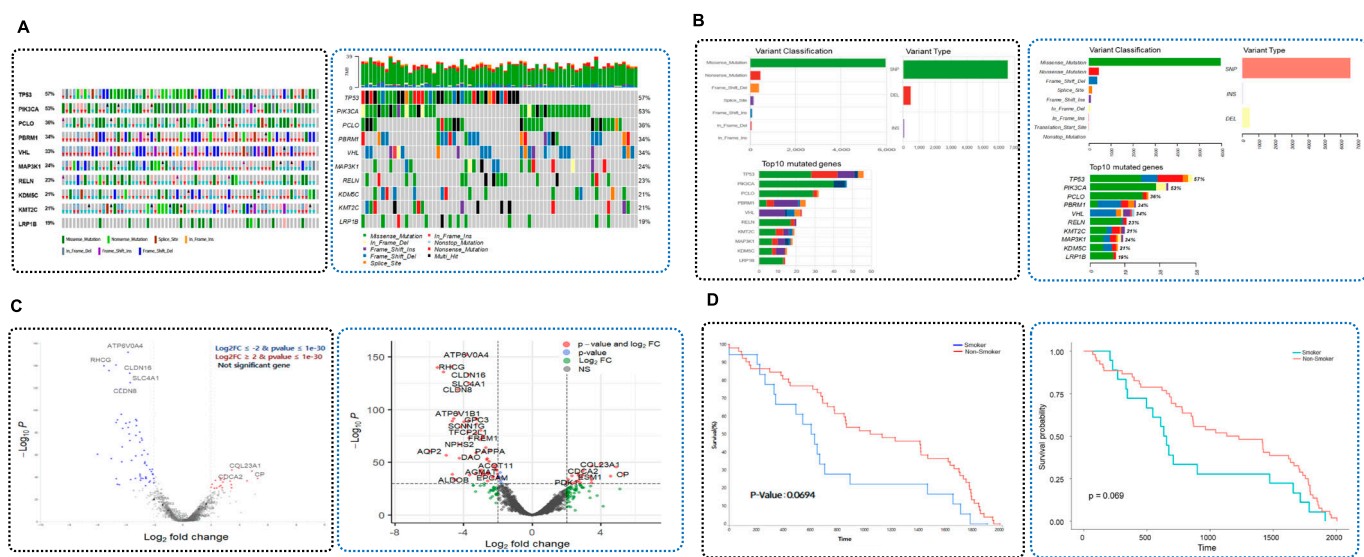

**Figure 6. Confirmation of K-CORE analytic tools.**
**(A, B, C, D)** OncoPrint plots showing mutation landscapes of selected genes. The left panel was visualized using the K-CORE analytic tool, and the right panel was generated using maftools in R; (B) summary plots of variant classification, variant types, and top 10 mutated genes. The left panels were generated using the K-CORE analytic tool, and the right panels were produced using maftools in R; (C) volcano plots displaying differential gene expression analysis. The left plot was produced using K-CORE, whereas the right plot was generated using R; and (D) Kaplan–Meier survival analysis comparing smokers and nonsmokers. Left panel: survival curves generated by K-CORE; right panel: survival curves produced using R.

page, example-data modules, user upload workflows, and major visualization pages. These design elements were introduced to reduce barriers for non–Korean-speaking users and first-time users. In addition, an English-language user manual has now been added to the portal through the customer-support section (Fig S3), although some instructional videos remain primarily available in Korean.

### Input schema, quality control, and operational constraints
Multidata projects require a clinical information file and at least one omics dataset. Supported upload types include DNA mutation, CNV, DNA methylation, RNA expression, proteome, phosphorylation, and fusion tables in a tab-delimited format. Sample IDs are used as the primary key for cross-modal matching. In the current portal configuration, the clinical upload table accepts up to 15 clinical variables, including the required outcome fields for recurrence and survival analysis (sample_id, rlps_yn, rlps_cnfr_drtn, death_yn, and death_cnfr_drtn). During upload, data type–specific help dialogs distinguish required and optional fields and provide downloadable sample templates for supported data types (Fig S2).

The upload QC workflow validates mandatory columns, value types, row-level formatting errors, and sample ID consistency before data are committed to a project. Incorrect values are flagged before analysis, and only visualization modules compatible with the successfully uploaded data types are activated. The current implementation is optimized for curated table-based inputs and hg38-centered genomic coordinates rather than arbitrary free-form files.

### Development of statistical and visualization tools

The K-CORE portal provides a comprehensive range of visualization tools to support diverse clinico-omics studies. With a focus on the integrated analysis of omics and clinical data, K-CORE suggests various statistical functions essential for precision medicine. The development and enhancement of these analytical functions were informed by insights gathered from clinical experts, bioinformatics professionals, and researchers. The implemented tools are user-friendly, versatile, and widely applicable, ensuring accessibility and adaptability to a broad spectrum of precision oncology research requirements. All genomic analyses were based on the hg38 reference genome, ensuring consistency and accuracy of variant interpretation.

### Omics–clinical integration capabilities

The K-CORE portal was designed to integrate and analyze multi-omics and clinical data. The uploaded multi-omics datasets and clinical information were matched and integrated based on each sample ID. The integrated multi-omics and clinical information datasets can be classified and filtered according to clinical parameters, enabling more targeted and precise analyses. This clinico-omics integration functionality not only enhances the exploration of omics data but also ensures that relevant clinical metadata are incorporated into statistical analyses, thereby improving the interpretability and translational relevance of research findings.

## Multi-omics landscape

To facilitate the integrated analysis of various biological alterations (e.g., mutations, CNVs, and fusions), the K-CORE portal incorporates OncoPrint and Circos plots. OncoPrint, based on the cBioPortal API, was modified to visualize genomic mutations, CNVs, and RNA and protein expression data, along with annotated clinical information. For variability in the preprocessing method of uploaded data among researchers, users can customize the filtering criteria to refine their analyses. Circos plot, one of the most widely used visualization tools, was implemented using CircosJS v2.2.0. It is dynamically customized based on the type of multi-omics data uploaded by the user, ensuring flexibility and adaptability to diverse research needs.

## Genomic mutation analysis

Genomic mutations are a fundamental aspect of bioinformatics analysis. The K-CORE portal offers a variant summary function; by selecting the genes of interest, users can visualize their frequencies and variant classifications. To support more analysis, a lollipop plot was developed using cBioPortal API, which enables precise visualization of mutated gene and protein phosphorylated position frequencies, providing detailed insights into mutation distribution at the genome level.

## CNV analysis

A CNV analysis function was developed based on the IGV API. The CNV tool visualizes altered gene IDs, chromosomal positions, and copy-number information from user data. It also provides gene- and position-based search options, enabling users to explore CNVs with enhanced visualization capabilities.

## Gene fusion analysis

The functionality of gene fusion analysis in the K-CORE portal was developed based on the hg38 reference genome and allowed users to examine fusion data within cohorts stratified by clinical information. Gene fusions were matched using gene transcript IDs, and the exact fusion breakpoints were visualized at the exon and intron levels to provide detailed insights into gene fusions.

## Expression profile analysis

The K-CORE portal suggests various analytical tools for expression data (e.g., RNA, Proteomics) and gene up/down-regulation research. Heatmap visualization based on CanvasXpress v43.9 supports not only expression data but also methylation and phosphorylation analyses. The heatmap includes clustering functionalities such as hierarchical clustering and K-means clustering, enabling researchers to identify patterns within their data. In addition, clinical information can be annotated alongside an expression heatmap for an overall understanding of the clinico-omics data. In addition, K-CORE provides box plots based on D3.js v7.8.5 for gene-specific group comparisons. Volcano plots, also developed using D3.js v7.8.5, facilitated differentially expressed gene (DEG) analysis between the selected control and comparison groups. DEG analysis in K-CORE was performed using the edgeR package (RRID:SCR_012802), one of the most widely used tools for DEG analysis. By default, the trimmed mean of the M-values normalization process was applied, and DEGs were computed based on user-customized groups. The results were visualized in a volcano plot, allowing researchers to efficiently interpret the expression differences between groups.

## Survival analysis

Real-time Kaplan–Meier (KM) survival analysis in the K-CORE portal was conducted using the KM and lifeline Python packages. Survival analysis provides not only overall survival but also recurrence and Cox regression analyses. Users can customize groupings based on their selected parameters (e.g., genomic variants and expression profiles), allowing for flexible and tailored analysis.

## Additional analytical tools

The K-CORE portal supports additional analytical tools to enhance the usability of visualization features. To facilitate variant annotation, a Variant Call Format-to-Mutation Annotation Format (MAF) converter (Variant Call Format-to-MAF converter) based on the vcf2maf package was provided. In addition, the MAF Merger function allows simultaneous merging of multiple MAF files. The RefVer converter was used to ensure reference genome compatibility and alignment with the K-CORE reference genome (hg38). Finally, to support broader research applications, principal component analysis (PCA) and correlation analysis tools are provided, allowing users to explore data structures and relationships more effectively.

## Validation of analytical results

The platform was assessed through three complementary lenses: task-based comparison with existing portals, analytical concordance, and representative upload-workflow timing in the deployed environment. Synthetic datasets were designed to mimic real-world genomic data distributions by anonymizing and randomly modifying values from the publicly available LinkedOmics dataset (Fig S7). These datasets were used exclusively for portal validation and demonstration.

Task-based comparison with existing portals is summarized in Table 1 and focuses on whether a platform can accommodate user-uploaded clinical and multi-omics data, support sample-level integration, and expose interactive analyses without local installation. To estimate operational timing under a realistic user submission, we prepared a representative multidata upload dataset comprising 100 clinical profiles together with 11,190 mutation entries, 294,210 CNV records, 309,500 methylation measurements, 460,500 RNA measurements, 425,100 proteome measurements, 2,422,600 phosphorylation records, and 93 fusion events, corresponding to ~3.9 million omics rows across seven omics layers. After submitting this dataset to the deployed portal, we measured the time required for upload and QC; for this 100-sample submission, the user upload and QC step completed in

~2 min. Because end-to-end response time depends on data composition, activated modules, and concurrent server load, this value should be interpreted as representative operational timing rather than a universal throughput limit.

Analytical validation focused on side-by-side comparison of K-CORE outputs with widely used R-based workflows, including maftools (RRID:SCR_024519) and edgeR (RRID:SCR_012802), to assess consistency and reproducibility (Figs 6 and S8). Survival outputs were also compared with the R-based Kaplan–Meier analysis.

# Data Availability

The source code for the core visualization modules of the K-CORE platform is publicly available on GitHub (https://github.com/K-CORE-NCDC). The repository includes documentation and example-data configurations to support reproducibility and implementation. Internal server-side scripts and any sensitive or confidential components have been excluded from the public release.

### Ethics statement

This study used publicly available and synthetic datasets for platform validation and did not involve the enrollment of human participants or intervention with human subjects.

# Supplementary Information

# Acknowledgements

We thank the users and collaborators who provided feedback during the development and testing of the K-CORE portal. This research was supported by the Healthcare Big Data showcase project, initiated by the Ministry of Health and Welfare (Grant Numbers: NCC-1902620, NCC-2002650, and NCC-2102860), and by a grant from the National Cancer Center funded by the Korean government, Republic of Korea (NCC-2310290).

### Author Contributions

J Hwang: data curation, software, validation, visualization, and writing—original draft, review, and editing.
JW Ahn: data curation, software, validation, visualization, and writing—original draft, review, and editing.
JW Lee: conceptualization, software, visualization, and writing—review and editing.
S-Y Jung: data curation, validation, and writing—review and editing.
E-G Lee: data curation, validation, and writing—review and editing.
H Chae: data curation, validation, and writing—review and editing.
H Koo: software and writing—review and editing.
H Cha: conceptualization and writing—review and editing.
J Kim: conceptualization and writing—review and editing.
K Kim: data curation, software, visualization, and writing—original draft, review, and editing.
D Lee: data curation, software, visualization, and writing—review and editing.
J Park: software, visualization, and writing—review and editing.
S-Y Kong: conceptualization, data curation, validation, and writing—review and editing.
KS Choi: conceptualization and writing—review and editing.
H-J Kim: conceptualization, data curation, software, supervision, funding acquisition, validation, visualization, and writing—review and editing.

### Conflict of Interest Statement

The authors declare that they have no conflict of interest.

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
