## [Reviewer comments · Life Science Alliance]

Development of K-CORE: A Web-Based Platform for Integrated Clinico-Genomic Analysis

Juyeon Hwang, JaeWoo Ahn, Jae Lee, So-Youn Jung, Eun-Gyeong Lee, Heejung Chae, Harim Koo, Hyosoung Cha, Junetae Kim, Kwangmin Kim, Dongwoo Lee, Junhyung Park, Sun-Young KONG, Kui Choi, and Hyun-Jin Kim

DOI: <https://doi.org/10.26508/lsa.202603659>

Corresponding author(s): Hyun-Jin Kim, National Cancer Center

Review Timeline:	Submission Date:	2026-02-09
	Editorial Decision:	2026-02-11
	Revision Received:	2026-04-06
	Editorial Decision:	2026-05-04
	Revision Received:	2026-05-12
	Accepted:	2026-05-12

Scientific Editor: Tim Fessenden

Transaction Report:

Please note that the manuscript was previously reviewed at another journal and the reports were taken into account in the decision-making process at *Life Science Alliance*. Since the original reviews are not subject to Life Science Alliance's transparent review process policy, the reports and author response cannot be published.

February 11, 2026

Re: Life Science Alliance manuscript #LSA-2026-03659-T

Hyun-Jin Kim
National Cancer Center

Dear Dr. Kim,

Thank you for transferring your manuscript entitled "Development of K-CORE: A Web-Based Platform for Integrated Clinico-Genomic Analysis" to Life Science Alliance. We are glad to consider publication of this resource describing a novel web platform developed by your group. In accordance with the offer that was conveyed by our colleagues at another journal, we invite you to submit a revised manuscript.

As we indicated, we will return the revised manuscript to reviewers for their evaluation. In case you have any questions or concerns I would be happy to discuss the revision in more detail via email or phone/videoconferencing. Please let me know which option you prefer, if any.

While you are revising your manuscript, please also attend to the below editorial points to help expedite the publication of your manuscript. Please direct any editorial questions to the journal office. When submitting the revision, please include a letter addressing the reviewers' comments point by point.

Thank you for this interesting contribution to Life Science Alliance. We are looking forward to receiving your revised manuscript.

Sincerely,

- A letter addressing the reviewers' comments point by point.
- An editable version of the final text (.DOC or .DOCX) is needed for copyediting (no PDFs).
- High-resolution figure, supplementary figure and video files uploaded as individual files: See our detailed guidelines for preparing your production-ready images, <https://www.life-science-alliance.org/authors>
- Summary blurb (enter in submission system): A short text summarizing in a single sentence the study (max. 200 characters including spaces). This text is used in conjunction with the titles of papers, hence should be informative and complementary to the title and running title. It should describe the context and significance of the findings for a general readership; it should be written in the present tense and refer to the work in the third person. Author names should not be mentioned.
- By submitting a revision, you attest that you are aware of our payment policies found here: <https://www.life-science-alliance.org/copyright-license-fee>

B. MANUSCRIPT ORGANIZATION AND FORMATTING:

May 4, 2026

RE: Life Science Alliance Manuscript #LSA-2026-03659-TR

Dr. Hyun-Jin Kim
National Cancer Center
323 Ilsan-ro
Ilsandong-gu
Goyang-si 10408
Korea, Republic of (South Korea)

Dear Dr. Kim,

Thank you for submitting your revised manuscript entitled "Development of K-CORE: A Web-Based Platform for Integrated Clinico-Genomic Analysis". We returned this work to the original Reviewer 2 whose comments are below. We would be happy to publish your paper in Life Science Alliance pending final revisions necessary to meet our formatting guidelines.

MANUSCRIPT ORGANIZATION AND FORMATTING:

To avoid unnecessary delays in the acceptance and publication of your paper, please read the following information carefully. Full guidelines are available on our Instructions for Authors page, <https://www.life-science-alliance.org/authors>

- Please consult our manuscript preparation guidelines <https://www.life-science-alliance.org/manuscript-prep> and make sure your manuscript sections are in the correct order.
- Please add an ORCID ID for the corresponding (and secondary corresponding) authors--you should have received instructions on how to do so.
- Please add a Summary Blurb/Alternate Abstract in our system.
- Please incorporate any points from the Conclusion section into the Discussion, as we only allow a Discussion section.
- Please add your main, supplementary figure, and table legends to the main manuscript text after the references section.
- Please add an Approval statement to your main manuscript text.
- Please add an Acknowledgements section to your main manuscript text
- In the reference list, citations should be listed with the authors' surnames and initials inverted. Where there are more than 10 authors on a paper, the first 10 will be listed, followed by 'et al.'
- Please add callouts for Figure 5 (A,B,C,D), Figure S3, Figure S4(A,B,C,D,E), Figure S6 (A,B,C), and Figure S7 (A,B) to your main manuscript text.

We welcome submissions of potential cover images for the issue of LSA in which your work would appear. If you have high quality images associated with this work, please feel free to email these, with a caption, to the journal office.

LSA encourages authors to provide a 30-60 second video where the study is briefly explained. These videos will be appear embedded with the manuscript online at Life Science Alliance, and on social media to promote the published paper and authors (for examples, see <https://docs.google.com/document/d/1-UWCfbE4pGcDdcgzcmiuJl2XMBJnxKYeqRvLLrLSo8s/edit?usp=sharing>). Corresponding or first-authors are welcome to submit the video. Please submit only one video per manuscript. The video can be emailed to contact@life-science-alliance.org

FINAL FILES:

The following items are required for acceptance.

The license to publish form must be signed before your manuscript can be sent to production. A link to the license to publish form will be available to the corresponding author only. Please take a moment to check your funder requirements.

Thank you for your attention to these final processing requirements. Please revise and format the manuscript and upload materials as soon as you are able.

Thank you for this interesting contribution to the literature. We look forward to publishing your paper in Life Science Alliance.

Sincerely,

May 12, 2026

RE: Life Science Alliance Manuscript #LSA-2026-03659-TRR

Dr. Hyun-Jin Kim
National Cancer Center
323 Ilsan-ro
Ilsandong-gu
Goyang-si 10408
Korea, Republic of (South Korea)

Dear Dr. Kim,

Thank you for submitting your Resource entitled "Development of K-CORE: A Web-Based Platform for Integrated Clinico-Genomic Analysis". It is a pleasure to let you know that your manuscript is now accepted for publication in Life Science Alliance. Congratulations on this interesting work.

Your article will publish open access upon publication under a CC-BY license.

DISTRIBUTION OF MATERIALS:

Again, congratulations on a very nice paper. I hope you found the review process to be constructive and are pleased with how the manuscript was handled editorially. We look forward to future exciting submissions from your lab.

Sincerely,
